# HCC in the Era of Direct-Acting Antiviral Agents (DAAs): Surgical and Other Curative or Palliative Strategies in the Elderly

**DOI:** 10.3390/cancers13123025

**Published:** 2021-06-17

**Authors:** Stefania Brozzetti, Marsia Tancredi, Simone Bini, Chiara De Lucia, Jessica Antimi, Chiara D’Alterio, Giuseppe Maria De Sanctis, Caterina Furlan, Vittoria Carolina Malpassuti, Pierleone Lucatelli, Michele Di Martino, Mario Bezzi, Antonio Ciardi, Rosa Maria Pascale

**Affiliations:** 1Department of Surgery “Pietro Valdoni”, Policlinico Umberto I, University of Rome La Sapienza, 00161 Rome, Italy; stefania.brozzetti@uniroma1.it (S.B.); marsiatancredi.1995@gmail.com (M.T.); chiara.delucia@uniroma1.it (C.D.L.); antimi.1732924@studenti.uniroma1.it (J.A.); dalteriochiara@gmail.com (C.D.); 2Department of Translational and Precision Medicine, Policlinico Umberto I, University of Rome La Sapienza, 00161 Rome, Italy; 3Department of Tropical and Infectious Diseases, Policlinico Umberto I, University of Rome La Sapienza, 00161 Rome, Italy; giuseppe.desanctis@uniroma1.it (G.M.D.S.); caterina.furlan@uniroma1.it (C.F.); 4Department of Statistical Sciences, University of Rome La Sapienza, 00161 Rome, Italy; vittoriacarolina.malpassuti@uniroma1.it; 5Department of Radiological Sciences Policlinico Umberto I, University of Rome La Sapienza, 00161 Rome, Italy; pierleone.lucatelli@gmail.com (P.L.); michele.dimartino@uniroma1.it (M.D.M.); mario.bezzi@uniroma1.it (M.B.); 6Department of Radiological, Oncological, Pathological Sciences, Policlinico Umberto I, Sapienza University of Rome, 00161 Rome, Italy; antonio.ciardi@uniroma1.it; 7Department of Medical, Surgery and Experimental Sciences, Division of Experimental Pathology and Oncology, University of Sassari, 07100 Sassari, Italy; patsper@uniss.it

**Keywords:** hepatocellular carcinoma, hepatitis C virus, HCV, HCC, HCC recurrence, DAA, surgery, locoregional therapy

## Abstract

**Simple Summary:**

This study investigated HCC onset in a cohort of patients receiving DAA therapy for HCV infection. It highlights that HCC onset after HCV-clearance might show more aggressive behavior and might exclude the patient from curative treatments, such as surgery or radiofrequency ablation. Moreover, HCC may develop in livers with mild to moderate fibrosis, indicating that multiple factors (host immune response, host metabolism, etc.) may play an important role in determining the cancer onset or its recurrence after HCV eradication. It is crucial to classify long-term chronically infected patients as at high risk for HCC development and to implement strict follow-up for them after eradication.

**Abstract:**

Hepatocellular carcinoma (HCC) accounts for 75–85% of primary liver malignancies, and elderlies have the highest incidence rates. Direct-acting antiviral agents (DAAs) have shown satisfying results in terms of HCV sustained viral response (SVR). However, data regarding HCC risk post-DAA-SVR is still conflicting. This study aims to consider HCC onset in moderate underlying liver disease. We conducted a retrospective study on 227 chronically infected patients (cHCV), treated with DAAs. Patients were divided into three groups: “*de novo occurrent HCC*”, “*recurrent HCC*”, and “*without HCC*”. Fifty-six patients aged <65 years (yDAA) were studied separately. HCC patients aged ≥65 years (DAA-HCC) were compared to a historical group of 100 elderly HCC patients, treated with peginterferon (Peg-IFN) ± ribavirin antiviral agents, non-SVR (hHCC). The HCC prevalence in DAA patients was 32.75%: “de novo occurrent’’ 18.13% and “recurrent’’ 14.62%, despite 42.85% of them having no fibrosis to mild or moderate fibrosis (F0-F1-F2). yDAA showed 5.36% “de novo occurrent” HCC. Curative procedure rates were compared between DAA-HCC and hHCC at the first and at recurrent presentation (22 (39.29%) vs. 72 (72%); 17 (30.36%) vs. 70 (70%), respectively (*p* < 0.001)). No significant difference was found in 3-year OS (*p* = 0.6). However, in cause-specific mortality analysis, HCC-related death was higher in the DAA-treated group, whereas cirrhosis-related death was more common in the historical group (*p* = 0.0288), considering together the two causes of death. A more accurate patient stratification according to multifactorial and new diagnostic investigations identifying HCC risk might allow an improvement in management and access to curative therapies.

## 1. Introduction

Hepatocellular carcinoma (HCC) is the fifth most common cancer in men and the seventh most common cancer in women. It is also the second leading cause of cancer death worldwide [1,2].

More than 70% of cases are due to chronic hepatitis related to hepatitis B virus (HBV) in the Asia Pacific region and sub-Saharan Africa and to hepatitis C virus (HCV) in Western countries [3,4,5].

The annual risk of developing HCC is about 3–3.5% in the general population in advanced liver disease [6]. Its prevalence, morbidity, and mortality, associated with low rates of diagnosis and treatment, have become one of the major public health challenges.

Antiviral agents that eliminate hepatitis C virus infection may prevent or reduce HCC risk in the liver with advanced fibrosis or cirrhosis. Until 2011, chronic hepatitis C therapy was limited to the combination of pegylated interferon (α2a or α2b) and ribavirin with a 30–75% eradication chance, depending on the viral genotype, lower for genotype 1 [7,8]. The introduction of interferon-free direct-acting antiviral (DAA) treatments, with the initial approval of sofosbuvir in December 2013, has changed the landscape of HCV therapy, and rates of sustained viral response (SVR) of more than 95% have been reported. These new drugs are different in terms of efficacy, side effects, and genotypic drug resistance. In 2015, over 100 million patients, the majority with liver cirrhosis, were treated with new antiviral drugs [9,10].

In Italy, F0-F1 patients with extrahepatic diseases (type 2 diabetes, kidney damage, immune-mediated manifestations, cardiovascular disease, obesity) have been included in the AIFA (Italian Medicines Agency) guidelines for DAA therapy since 2017 [11].

In this new scenario, the incidence of HCV infection, the evolution in cirrhosis, and the incidence of HCC should have decreased dramatically over a few decades. Instead, unexpected data from recent studies showed that SVR does not eliminate the risk of developing HCC post-DAA. Other studies raised the suspicion that DAAs might promote an early HCC occurrence (de novo) or recurrence in cured HCC patients [12,13,14,15]. This study aims to show the impact of HCC after new antiviral therapies in our cohort of patients affected by different features of liver damage and to compare treatments and outcomes with a historical group of HCC patients. Finally, the data gained from the study are discussed to clarify the suspicion of a vulnerability in HCC onset as a result of multiple involved pathways emerging.

## 2. Methods

### 2.1. Study Setting and Participants

Retrospective data of 227 patients treated with DAAs from October 2015 to December 2019 were collected and analyzed. We included adult patients (>18 years of age) with cHCV infection. HCV infection was confirmed with quantitative real-time polymerase chain reaction (RT-qPCR) and genotyping/subgenotyping was carried out. Exclusion criteria were HIV/HBV coinfection, alcohol abuse, not achieving sustained viral response, therapy drop-out, liver transplantation, and a follow-up of less than 12 months. A flow chart of the patients’ enrollment is shown in Figure 1.

At baseline, data on clinical characteristics and parameters, namely age, sex, anthropometric parameters, and comorbidities (ASA score), and extrahepatic diseases (type 2 diabetes, kidney damage, immune-mediated manifestations, cardiovascular disease, obesity) were collected.

Liver damage was assessed according to the Child–Turcotte–Pugh score (CTP); the Model for End-Stage Liver Disease (MELD) score and the liver stiffness scales (METAVIR scale, Fibroscan values, F4 value). Platelet count, splenomegaly, varices, and serum alpha-fetoprotein (AFP) levels pre-DAA were also considered.

HCCs were staged according to the Barcelona Clinic Liver Cancer (BCLC) staging system considering lesion number, size, and localization. DAA treatment regimens were recorded for all patients.

Liver imaging study (ultrasound (US), computed tomography/magnetic resonance (CT/MRI)) and serum alpha-fetoprotein (AFP) levels were requested to rule out HCC before starting antiviral therapy.

The liver fibrosis stage was assessed by histological exam in the surgical specimen or by external/intraoperative biopsy (METAVIR F0-F4) by noninvasive methods including transient elastography and laboratory determinations (FIB-4). Elastography, analyzed by the Fibroscan (kPa value), was categorized according to international recommendations. Cut-off values of 10 and 12.5 kPa were considered for F3 and F4, respectively. The FIB-4 was calculated using the following formula: (aspartate aminotransferase (AST; U/L) × age (years))/(platelet count (10^9^/L) × alanine aminotransferase (ALT; U/L)^1/2^). A cut-off value of 3.25 was considered an index of severe fibrosis [16,17].

The severity of cirrhosis was also defined by US, CT, and MRI images according to hepatic size and profile, nodularity, portal flow and diameter, esophageal varices, and splenomegaly [18]. Steatosis was detected by US/MRI or by liver biopsy or surgical sampling. The DAA regimen used was based on international guidelines [19]. The efficacy of antiviral therapy, referred to as SVR, was defined as undetectable HCV RNA (<15 IU/mL) tested at the end of DAA protocol, at 24 weeks, and at 1 year. Surveillance with AFP levels and US were performed at 6 and then every 12 months after DAA therapy. Any new liver nodule (≥5 mm) found on ultrasound was followed up by the US at 3 months and (≥10 mm) by CT scan or MRI with liver-specific contrast medium.

HCC (occurrent or recurrent) was staged according to the BCLC system, and nodules were assessed for size, localization, and number [20,21]. Patients were divided into three groups: HCC-O “de novo occurrent”, HCC-R “recurrent”, and patients without a history of HCC onset “without HCC”. The time from achieving SVR to the onset of HCC was also recorded and assessed. HCC treatment modality was established in a multidisciplinary expert team (hepatologists, surgeons, radiologists, and oncologists) according to the BCLC staging system combined with patient general conditions and liver damage. Surgical resection (SR), radiofrequency (RF) ablation, and liver transplantation (LT) (not available due to the patients’ age) were classified as potentially curative therapies, while transcatheter arterial chemoembolization (TACE), intra-arterial radioembolization (IART), and systemic therapy (ST) with different schemes were classified as noncurative treatments [22].

One hundred seventy-one patients (median age of 73 years (IQR: 67–81.)) were selected from this cohort, and those with HCC (HCC-O and HCC-R) after DAA therapy (DAA-HCC) were compared to a historical group of HCC patients (hHCC) (median age of 74.5 years (IQR: 70–79)) treated in a period before the era of DAA therapy with Peg-INF ± ribavirin, who did not achieve HCV SVR (except 4 patients). Eleven of the included hHCC patients had already undergone curative-specific treatments (10 radiofrequency ablations and 1 surgical resection). In the DAA-treated cohort, patients in the “*without HCC*” group did not have any previous history of HCC, whereas patients in the “HCC-R” group received curative treatment for the previous HCC lesion: 16 patients underwent surgical resection and 9 patients received radiofrequency ablation. Characteristics of the patients, tumors, modality of treatment, overall survival (OS), and cause-specific mortality were analyzed. Fifty-six DAA patients under 65 years of age were analyzed separately (Appendix A).

### 2.2. Statistical Analysis

According to the test of normality (Shapiro–Wilk test), mean (+/−standard deviation (SD)) or median (interquartile range (IQR)) were calculated for normally and non-normally distributed continuous data, respectively.

The continuous variables were assessed using the Student’s *t*-test for parametric variables or the Mann–Whitney U test for nonparametric variables.

Categorical variables were compared using the chi-square test or Fisher’s exact test.

Kaplan–Meier curves were generated to compare time from SVR attainment to diagnosis of HCC and HCC-free survival by treatment regimen among different groups.

A *p*-value < 0.05 was considered statistically significant.

The statistical analysis was carried out using the R software (version 3.6.1).

## 3. Results

The cohort of elderly patients treated with DAA therapy included 139 men and 88 women (ratio 1.6:1) with a median age of 68 years (IQR: 65–78). All patients achieved SVR. While 138 patients had been previously treated with interferon (Peg-INF ± ribavirin)-based therapies, 89 patients had never received antiviral drugs before (naive). The median follow-up after DAA treatment was 45 months (IQR: 24–52).

After DAA therapy, HCC prevalence was higher in the group of elderly patients: 56 (32.75%) out of 171, of which 31 (18.13%) had “de novo HCC occurrence” and 25 (14.62%) had “recurrent HCC”. However, only 3 cases of HCC occurred in the population aged <65 years (Appendix A). In the “without HCC” group, patients did not have any history of HCC before DAA therapy. Male sex in the elderly was more frequently associated with HCC development: 25 males vs. 6 females in HCC-O group, 18 males vs. 7 females in HCC-R group (*p* < 0.001), and 67 males vs. 33 females in the hHCC group (*p* < 0.001). Clinical, laboratory, and instrumental variables of 171 DAA-treated elderly patients are reported in Table 1. Chronic viral hepatitis has been the object of the most extensive efforts in grading and staging liver damage, stimulated by the advent of new antiviral therapies. According to the international guidelines, we combined invasive and noninvasive methods and biological tests to evaluate liver function and grade of fibrosis. There was no significant difference among the groups of DAA patients for the Child–Turcotte–Pugh (CTP) score, MELD score, METAVIR, FIB-4, liver stiffness, or platelet count. In HCC-O and HCC-R groups, a total of 53 patients (94.64%) had a CTP score A (5.6), 3 patients (5.36%) had a CTP score B (7, 8, 9), and no patients had a CTP score C. F0-F2 liver fibrosis was found in 40.53% of all DAA-treated elderly patients and in 42.86% of DAA-HCC (32.14% F0-F1). Noninvasive methods for parenchyma evaluation showed medians above the cut-off for the predictive value for severe fibrosis in HHC-O or HCC-R groups. The IQR also included values under the cut-off, due to the wide presence of patients showing mild fibrosis (Table 1). No patients were found to develop severe steatosis or steatohepatitis. HCV genotype was not associated with differences in HCC rates among the three groups. AFP serum values pre-DAA treatment were commonly higher in HHC-O and HCC-R groups in comparison with the “without HCC” group (4.5 (IQR: 2.8–9.32) vs. 4.6 (IQR: 3–6.4) vs. 2.4 (IQR: 1.12–4.11), *p* < 0.001); however, AFP level was never markedly elevated.

The median latency time from SVR to HCC detection was 14 months (IQR: 7.5–23.5) in HCC-O group vs. 13 months (IQR: 5–22) in HCC-R group (*p* = 0.287). Tumor characteristics and treatments are shown in Table 2. According to the BCLC system, 26 patients (46.4%) were in stage A, 22 (39.3%) were in stage B, and 8 (14.3%) were in stage C, without significant differences between HCC-O and HCC-R groups.

Tumor characteristics and treatment allocation to each stage between selected patients (DAA-HCC) and hHCC group are summarized in Table 3. hHCC patients had the advantage of a larger combination of curative treatments, resection and RF ablation, compared to DAA-HCC (72 (72.0%) vs. 22 (39.29%) and 70 (70%) vs. 17 (30.36%) at first and second tumor presentation, respectively; *p* < 0.001). The presence of more multinodular forms or larger HCC dimensions is reflected in the data found in the DAA HCC group. This evidence was one of the reasons that prompted us to start our study (Figure 2 and Figure 3).

Pathohistological diagnosis of resected/biopsied tumors resulted as “trabecular type”, except one in the hHCC group that resulted as combined “hepato-cholangiocarcinoma”. Morphological staging according to TNM classification, including resection margin assessment and grading, provided uniform data and similarly staged lesions among groups of resected patients.

In December 2020, none of the DAA-*HCC* patients had completed a five-year follow-up. The 3-year OS was 76.02% (IC 95%: 64.90–89.04) in DAA-HCC vs. 88.93% (IC 95%: 82.96–95.32) in h-HCC group (Table 3) (Figure 4). Cause-specific mortality rates are reported in Table 3. We found statistically significant different rates in HCC (7 (53.85%) vs. 13 (25%)) and cirrhosis (2 (15.38%) vs. 24 (46.15%)) between DAA-HCC and h-HCC group (*p* = 0.0288), considering only these two causes of death. Moreover, from December 2020, the end date of our retrospective study, to April 2021, three other elderly patients analyzed in the “without HCC” group developed HCC. The clinical characteristics of 56 patients less than 65 years old are summarized in Appendix A. Only three patients developed HCC, and one liver transplantation (LT), one liver resection, and one transcatheter arterial chemoembolization (TACE) for a multifocal disease were administered.

## 4. Discussion

Hepatitis C virus (HCV) infection is the main leading etiology of hepatocellular carcinoma in Europe, North America, and Japan. Emerging risk factors for HCC development such as NASH, type 2 diabetes mellitus, and obesity may determine a synergistic or an independent risk [23].

HCV antiviral treatment with IFN–ribavirin therapy produced an SVR in 30–75% of patients with significant differences according to genotypes (with lower effect in type 1 HCV with a higher level of viremia) [7,8]. Difficult tolerability for the severity of side effects and the length of treatment (24–48 weeks) led to low adherence. Infection relapse following an initial sustained virological response was common, leading to therapeutic failure. On the other hand, first and second generations of direct-acting antiviral agents (DAAs) proved to be safe, well-tolerated, of short and easy administration, and highly effective in the eradication of HCV (more than 95% of cases) [9,10].

However, interferon (IFN)-mediated HCV eradication reduces but does not eliminate the risk of HCC once cirrhosis has been established; in the same conditions, DAA therapy may also not eliminate the risk of HCC. Despite HCV clearance and improved liver function, the process induced by chronic HCV infection in patients over 20–40 years old, characterized by chronic hepatic inflammation and progressive liver fibrosis, determines the initiation of neoplastic transformation by irreversible somatic genetic or epigenetic alterations that establish the mechanism of HCC development that is not modified by DAA treatment [24,25,26]. Indeed, we observed an increase in HCC-related mortality and a decrease in cirrhosis-related mortality in the DAA-treated patients as compared to the historical IFN-treated group.

High fibrosis grade is also significantly associated with an elevated risk of HCC onset in patients that cleared HCV infection. *Conti* et al., in an observational study enrolling 34 cirrhotic patients who received DAA treatment, reported a 3.16% liver cancer rate, almost the same as in untreated chronically infected patients [13]. Surprisingly, the recurrence rate in a population with HCC cured by ablation or resection followed by HCV elimination was higher than that of the untreated population, 28.81% vs. 20% [13]. In a similar study, the research group of Reig et al. confirmed an incidence of 27.6% of HCC recurrence in patients completing the DAA treatment with an SVR [12]. Additionally, Cardoso et al. reported an HCC incidence of 7.4% in the first year after SVR from DAA treatment, higher than the previously reported incidence rate for IFN regimens (1.2–1.4%) [27,28]. Indeed, some other multicentric studies did not share a higher rate of HCC occurrence or recurrence in DAA treatment as compared to IFN-treated patients or untreated controls [29,30]. We observed increased rates of both HCC occurrence and recurrence in our DAA-treated elderly patients (18.13% and 14.62%, respectively) as compared to HCC occurrence and recurrence rates observed in untreated patients that range from 5 to 10% [31].

The vast majority of published studies missed focusing on the correlation between tumor characteristics, stage, and treatment decisions after DAA therapy compared to HCC in IFN-treated or untreated controls. Despite the availability of highly effective direct-acting antiviral agents, the morbidity and incidence of liver-related complications of HCV, including HCC, is likely to persist. The endpoint of DAA therapy is to cure HCV infection and to prevent the HCV-related liver and extrahepatic complications including advancement and decompensation of liver cirrhosis, HCC development, and HCV-related autoimmune diseases [32,33].

Our cohort of DAA-treated patients showed a higher incidence of de novo or recurrent HCC at median 14 and 13 months, respectively. In some cases, HCC onset was observed even after 5 years. Therefore, HCC surveillance in SVR responders should be maintained for at least 5 years, particularly in long-lasting HCV infection. Liver tumors after DAA often showed more aggressive behavior, were larger in size, and were more often multifocal than the ones in untreated or IFN patients, which could explain the less favorable treatments offered. Severe fibrosis/cirrhosis is broadly accepted as the pathogenetic mechanism for HCC development after SVR, with hundreds of publications available. However, the HCC development in the noncirrhotic liver after a cleared HCV infection is a reason for the discrepancy in expectations, particularly if it develops with multinodular or aggressive patterns, as detailed in some case reports published [34]. Since 2017, in Europe, DAA therapy has also been administered to the F0-F2 liver, in case of comorbidities. Although relatively less common (4.4–10.6%), liver cancer arises in noncirrhotic and HCV-positive patients [35,36]. In these cases, its prognosis is generally better than in cirrhosis, especially if HCC is at an early stage when curative therapy, liver resection, or radiofrequency ablation can be performed. In our cohort, some HCV-cleared patients showing F0-F2 fibrosis with no alcohol abuse or severe metabolic syndrome developed HCC with an aggressive pattern. The combination of this evidence on liver fibrosis (sometimes low), liver function (acceptable), and tumor stage (frequently advanced in size, number, or vascular invasion) (Table 3) supports the hypothesis that different and not previously considered factors may lead to tumor outgrowth after DAA treatment. The interaction of HCV with its human host is complex and multilayered. HCC development is probably determined by several viral and host factors. Direct and indirect mechanisms of HCV-induced HCC include activation of multiple host pathways such as liver fibrogenic pathways, cellular and survival pathways, and interaction with the immune and metabolic systems. Host factors also play a major role in HCV-induced HCC, as evidenced by genomic studies identifying polymorphisms in immune, metabolic, and growth signaling systems associated with increased risk of HCC [37,38,39].

The hypothesis raised by researchers points at considering that the HCV elimination may open a temporary immunosuppressive phase that might determine the ideal conditions for the growth of dormant micronodules [40]. In support of the hypothesis of DAA-induced immunosuppression, there was evidence of an increase in reactivations of the hepatitis B virus and herpes viruses after the new DAA therapies [41,42]. Our data agree that the greatest incidence of HCC among DAA-treated patients was in the first months after HCV eradication; therefore, a leading role of the immune system in determining rapid HCC evolution shall not be excluded. Patients treated with DAA shall be followed up strictly after HCC eradication, especially in the case of chronic HCV infection, considering that potentially transformed cells may proliferate in conditions of reduced immune surveillance.

Our study highlighted that HCC may develop in the noncirrhotic liver after SVR mediated by DAA treatment, sometimes showing more aggressive behavior. Unfortunately, this study has some limitations: first, it had a relatively short follow-up time as a consequence of the recent introduction of DAAs in Italy (2015), and further information about the HCC incidence after viral cure is still needed. Second, our patients had a diagnosis of HCV infection 15–30 years before the DAA regimen, and the development of HCC may be linked to the continuous viral damage also present in noncirrhotic patients. Since HCV-related HCC onset is time-dependent, this investigational hypothesis could suggest new studies comparing the onset of HCC in patients with a short-lasting HCV infection treated with DAAs. Third, it was not possible to introduce a control group that was not treated with antiviral therapy due to ethical restrictions. A further limitation is represented by the sample size of our population due to the fact that this is a single-center study.

## 5. Conclusions

Some aspects of the impact of DAA therapy on mechanisms of HCC onset are still debated. The previous case series considered cirrhosis as the driver of HCC development. Underground moderate fibrosis should induce the investigation of other pathophysiological mechanisms. HCV infection exposition-time and the immunological state after DAA therapy may play important roles. Our data suggest that DAA therapy should be prioritized in patients with newly diagnosed HCV infection in order to successfully prevent liver damage and HCC onset.

Data obtained from this limited cohort of patients have to be interpreted with caution, and larger numbers of patients have to be studied in order to identify the responsible pathway for HCC development in noncirrhotic patients achieving SVR.

The future is bright for HCV patients since newly diagnosed HCV patients have a short viral exposition due to the improved access to DAA therapy.

## Figures and Tables

**Figure 1 cancers-13-03025-f001:**
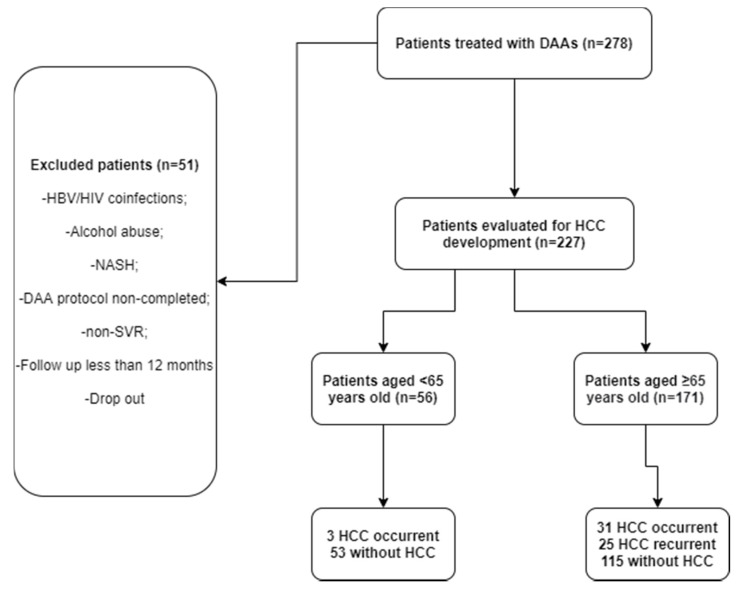
Flow chart of patient selection criteria in this retrospective study.

**Figure 2 cancers-13-03025-f002:**
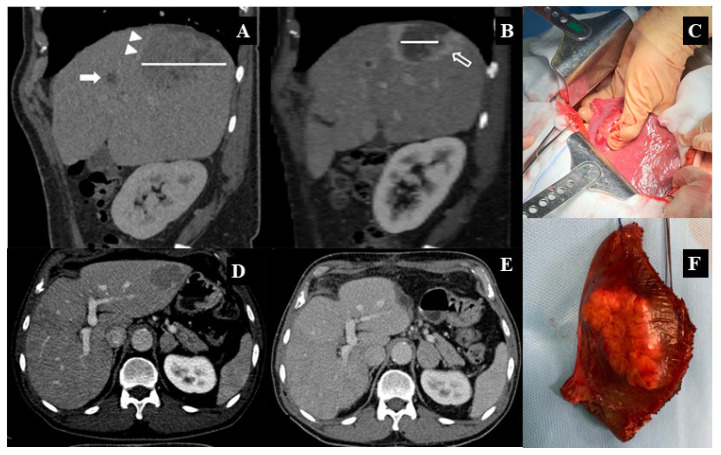
Intraoperative images compared to relative CT scans for one DAA-treated patient. (**A**) Sagittal reconstructed CT image during the delayed phase shows the huge HCC lesion in liver segment VII (9 cm: white line) with infiltration of right hepatic vein (arrowheads) and the tiny satellite nodule in the VIII° segment. (**B**) After radioembolization (SIRT), both lesions present central necrotic area and a fibrotic peripheral rim. The bigger lesion is also reduced in size (white line). (**D**) Axial CT image during the delayed phase shows another HCC lesion across the liver segments II° and III° in a noncirrhotic liver. (**C**) Intraoperative view. (**F**) Surgical specimen. (**E**) After surgery, a small fluid collection is appreciable along the resection margin, without any sign of HCC recurrence.

**Figure 3 cancers-13-03025-f003:**
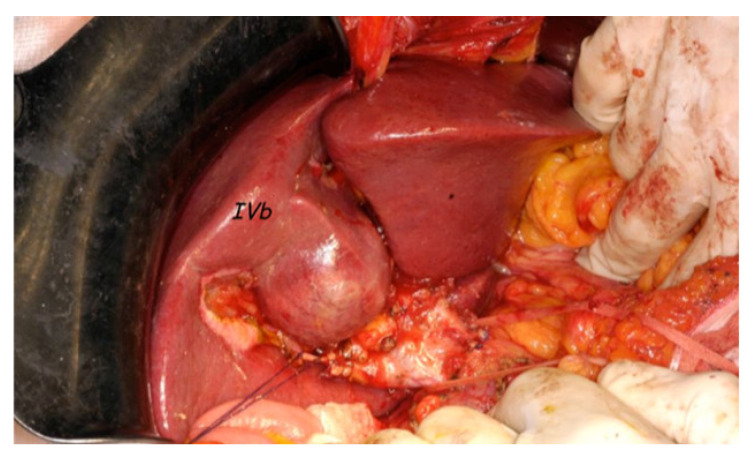
Intraoperative image of one IFN-treated patient. The HCC lesion in a noncirrhotic liver can be observed in segment IV.

**Figure 4 cancers-13-03025-f004:**
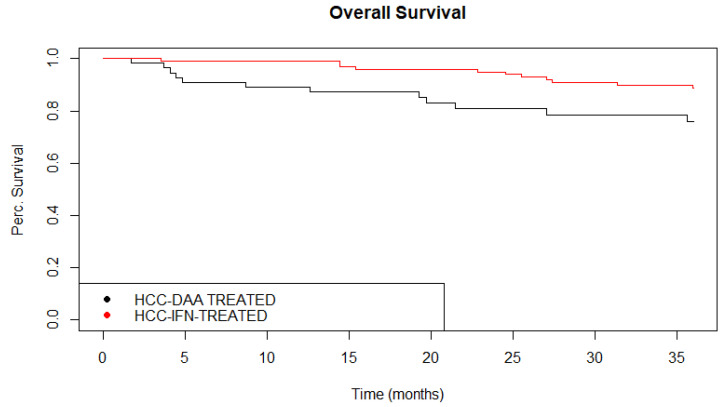
Kaplan–Meier 3-year overall survival in DAA-HCC group vs. hHCC group. Kaplan–Meier analysis of patients’ overall survival in historical IFN-treated group (non SVR) compared with overall survival of patients in DAA-treated group.

**Table 1 cancers-13-03025-t001:** Characteristics of the elderly population studied.

Variable	HCC-O (*n* = 31)	HCC-R (*n* = 25)	Without HCC (*n* = 115)	Total (*n* = 171)	*p* Value
Age, years, median (IQR)	73 (67–81.50)	69 (67–75)	74 (67–80)	73 (67–80)	0.315
Male, *n* (%)	25 (80.65)	18 (72)	54 (46.96)	97 (56.73)	<0.001
Female, *n* (%)	6 (19.35)	7 (28)	61 (53.04)	74 (43.27)	
BMI, kg/m^2^, median (IQR)	26.10 (25.5–26.5)	26.10 (26–26.5)	26 (25.5–26.8)	26 (25.5–26.7)	0.672
Comorbidities, *n* (%)					
Obesity	2 (6.45)	1 (4)	16 (13.91)	19 (11.11)	0.373
Diabetes mellitus 2	7 (22.58)	4 (16)	13 (11.3)	24 (14.04)	0.239
Metabolic syndrome	5 (16.13)	3 (12)	12 (10.43)	20 (11.17)	0.592
Chronic obstructive pulmonary diseases	3 (9.68)	2 (8)	4 (3.48)	9 (5.26)	0.203
Cardiovascular diseases	5 (16.13)	4 (16)	0 (0)	9 (5.26)	<0.001
Kidney failure	4 (12.9)	3 (12)	17 (14.78)	24 (14.04)	1
Hypertension	11 (35.48)	7 (28)	27 (23.48)	45 (26.32)	0.372
Hemoglobin disorders	2 (6.45)	2 (8)	5 (4.35)	9 (5.26)	0.666
Autoimmune diseases	3 (9.68)	2 (8)	2 (1.74)	7 (4.09)	0.053
ASA, *n* (%)					0.336
I	21 (67.74)	13 (52)	60 (52.17)	94 (54.97)	
II	9 (29.03)	11 (44)	55 (47.83)	75 (43.86)	
III	1 (3.23)	1 (4)	0 (0)	2 (1.17)	
HCV Genotype, *n* (%)					0.8693
1	19 (61.29)	17 (68)	73 (63.48)	109 (63.74)	
Non-1	12 (38.71)	8 (32)	42 (36.52)	62 (36.26)	
Child–Turcotte–Pugh score, *n* (%)					1
A	29 (93.55)		108 (93.91)	161 (94.15)	
B	2 (6.45)		7 (6.09)	10 (5.85)	
C	0 (0)		0 (0)	0 (0)	
MELD, median (IQR)	9 (7–11.5)	9 (7–11)	8 (7–9)	8 (7–10)	0.095
METAVIR, *n* (%)					0.385
F0-1	7 (22.58)	11 (44)	32 (27.83)	50 (29.24)	
F2	4 (12.90)	2 (8)	17 (14.78)	23 (13.45)	
F3	5 (16.13)	4 (16)	10 (8.70)	19 (11.11)	
F4	15 (48.39)	8 (32)	56 (48.70)	79 (46.20)	
FIB-4, median (IQR)	4 (2.34–5.83)	4.19 (2.95–6)	3.21 (1.74–6.17)	3.56 (1.95–6.08)	0.246
Stiffness, median (IQR)	12 (8.63–16.8)	10.4 (6–14)	12.5 (6.8–18.6)	11.8 (6.4–17.05)	0.218
Steatosis	2 (6.45)	2 (8)	26 (22.61)	30 (17.54)	0.055
Platelets, 10^9^/L, median (Range)	147 (42–362)	153 (32–464)	143 (29–350)	147 (29–464)	0.899
Varices, *n* (%)					0.072
0	22 (70.97)	12 (48)	84 (73.04)	118 (69.01)	
1	8 (25.81)	13 (52)	27 (23.48)	48 (28.07)	
2	1 (3.23)	0 (0)	4 (3.48)	5 (2.92)	
AFP pre-DAA, UI/L, median (IQR)	4.5 (2.8–9.32)	4.6 (3–6.4)	2.4 (1.12–4.11)	3 (1.45–5.05)	<0.001
DAA, *n* (%)					0.9164
SOF-Based	26 (83.87)	20 (80)	92 (80)	138 (80.70)	
Non-SOF-Based	5 (16.13)	5 (20)	23 (20)	33 (19.30)	

SOF, sofosbuvir; *p*-value was calculated between HCC-O, HCC-R, and cases without HCC.

**Table 2 cancers-13-03025-t002:** Main characteristics of de novo occurrent and recurrent HCCs.

Variable	HCC-O	HCC-R	TOTAL	*p* Value
Age, years, median (IQR)	73 (67–81.50)	69 (67–75)	70 (66.75–80)	0.271
Male, *n* (%)	25 (80.65)	18 (72)	43 (76.79)	0.657
Female, *n* (%)	6 (19.35)	7 (28)	13 (23.21)	
BMI, kg/m2, median (IQR)	26.10 (25.5–26.5)	26.10 (26–26.5)	26.10 (25.65–26.5)	0.409
HCV Genotype, *n* (%)				0.81
1	19 (61.29)	17 (68)	36 (64.29)	
Non-1	12 (38.71)	8 (32)	20 (35.71)	
DAA, *n* (%)				0.738
SOF-Based	26 (83.87)	20 (80)	46 (82.14)	
Non-SOF-Based	5 (16.13)	5 (20)	10 (17.86)	
Onset from SVR (months), median (IQR), (Range)	14 (7.5–23.5)	13 (5–22)	14 (7–23.25)	0.287
	(3–57)	(1–41)	(1–57)	
Numbers of nodules, *n* (%)				0.631
1	11 (35.48)	12 (48)	23 (41.07)	
2	6 (19.35)	3 (12)	9 (16.07)	
3	14 (45.16)	10 (40)	24 (42.86)	
HCC size (cm), median, (Range)	2.8 (1–8)	2.2 (0.8–5.1)	2.6 (0.8–8)	0.225
Site, *n* (%)				0.958
Right liver	21 (67.74)	18 (72)	39 (69.64)	
Left liver	10 (32. 26)	7 (28)	17 (30.36)	
BCLC STAGE, *n* (%)				0.874
0-A	13 (41.935)	12 (48)	25 (44.64)	
B	13 (41.935)	9 (36)	22 (39.29)	
C	5 (16.13)	4 (16)	9 (16.07)	
I Treatment, *n* (%)				1
Curative treatments	12 (38.71)	10 (40)	22 (39.29)	
Palliative treatments	19 (61.29)	15 (60)	34 (60.71)	
II Treatments, *n* (%)	8 (25.81)	9 (36)	17 (30.36)	0.594
Curative treatments	2 (25)	2 (22.2)	4 (23.53)	
Palliative treatments	6 (75)	7 (77.8)	13 (76.47)	
III treatments, *n* (%)	2 (6.45)	3 (12)	5 (8.93)	0.647
Curative treatments	0 (0)	1 (33.33)	1 (20)	
Palliative treatments	2 (100)	2 (66.67)	4 (80)	

*p*-value was calculated comparing HCC-O and HCC-R groups.

**Table 3 cancers-13-03025-t003:** Comparison between DAA-HCC and h-HCC groups.

Variable	DAA-HCC (*n* = 56)	h-HCC (*n* = 100)	*p* Value
Age, years, median (IQR)	70 (66.75–80)	74.50 (70–79)	0.064
Male, *n* (%)	43 (76.69)	67 (67)	0.270
Female, *n* (%)	13 (23.21)	33 (33)	
BMI, Kg/m2, median (IQR)	26.10 (25.65–26.5)	27 (26–28)	<0.001
Comorbidities, *n* (%)			
Hypertension	7 (12.5)	71 (71)	<0.001
Cardiovascular diseases	5 (8.93)	19 (19)	0.149
Pulmonary diseases	5 (8.93)	41 (41)	<0.001
Kidney failure	9 (16.07)	1 (1)	<0.001
Diabetes mellitus 2	11 (19.64)	20 (20)	1
Metabolic syndrome	8 (14.26)	2 (2)	0.004
Malnutrition	0 (0)	3 (3)	0.553
Characteristics of nodules			0.035
Single	23 (41.07)	60 (60)	
Multiple	33 (58.93)	40 (40)	
HCC size (cm), median, (Range)	2.6 (0.8–8)	2.65 (1–12.3)	0.629
Site, *n* (%)			0.433
Right liver	39 (69.64)	62 (62)	
Left liver	17 (30.36)	38 (38)	
I Treatment, *n* (%)			<0.001
Curative treatments	22 (39.29)	72 (72)	
Palliative treatments	34 (60.71)	28 (28)	
II treatment, *n* (%)	17 (30.36)	70 (70)	<0.001
Curative treatments	4 (23.53)	40 (57.14)	0.027
Palliative treatments	13 (76.47)	30 (42.86)	
3-year OS, survival %	76.02	88.93	0.6
(IC 95%)	(64.90–89.04)	(82.96–95.32)	
Death, *n* (%)	13 (23.21)	52 (52)	0.008
Cause of death, *n* (%)			
HCC	7 (53.85)	13 (25)	0.089
Liver disease/Cirrhosis	2 (15.38)	24 (46.15)	0.059
Other	4 (30.77)	15 (28.85)	1
HCC vs. Liver disease/Cirrhosis, *n* (%)			0.0288
HCC	7 (77.78)	13 (35.14)	
Liver disease/Cirrhosis	2 (22.22)	24 (64.86)	

*p*-value was calculated between DAA-HCC and h-HCC groups.

## Data Availability

Available at request to the corresponding author.

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
