# Peer review of "HCC in the Era of Direct-Acting Antiviral Agents (DAAs): Surgical and Other Curative or Palliative Strategies in the Elderly"

_cancers, 2021, doi:10.3390/cancers13123025_

Round 1

Reviewer 1 Report

This is an interesting project.

major issues:

  • the statistical model fails to take into account the baseline differences.
  • the historical control should be matched to the DAA patient group

minor issues : - Table 2 has a header typo

Author Response

Thank you for your comments. 

Statistical analisis was corrected, unfortunately it was not possible to match by age, sex and BMI the two groups, however, we believe that our results might be interesting as well. 

English language was extensively revised, as well as the methods section and results. 

Reviewer 2 Report

The Authors investigated HCC onset in a cohort of patients receiving DAA therapy for HCV infection. They show that HCCs after HCV-negativization show features of a more aggressive behaviour. Finally, they discuss that given that HCC may develop in livers with mild to moderate fibrosis they suggest that multiple factors (host immune response, host metabolism etc) may play an important role in determining the cancer onset or its recurrence after HCV negativization. 

The Authors should indicate if there are any differences in term of HCC onset between cases undergoing DAA therapy naive and cases undergoing DAA therapy after other treatments i.e. Interferon (Peg-INF ± Ribavirin)-based therapies.

Author Response

Thank you for your comments. 

English language was extensively revised, as well as the methods section and results. No differences were found in HCC onset in naive patients vs patients submitted to previous IFN-based therapies.  

Reviewer 3 Report

the topic of the paper has been largely debated in last years.

In my opinon there is a huge bias in the study design and methodology. HCC recurrence should be evaluated among patients with a history of previous HCC and the recurrence rate cannot be calculated on the overall population (including patients with and without a history of HCC, without any distinction). I think the analysis should be done keeping in mind that there should be 2 groups of patients (with and without previous HCC, for having recurrence rate and de novo rate of HCC), or removing patients with previous HCC from the overall population and evaluating only occurrence HCC rate.

Author Response

(The authors gave the same response as above.)

Round 2

Reviewer 3 Report

There is still a huge bias.

The HCC recurrence rate should be analysed evaluating only the cohort of patients with a previous history of HCC and not the whole enrolled cohort of patients.

Author Response

Thank you for your comments, statistical analysis have been revisioned, we also specified that we calculated HCC prevalence. 

In the Without HCC group enrolled patients have no history of HCC, only patients in the HCC-R group have history of HCC before DAA therapy.